# Ambipolar to Unipolar Conversion in C_70_/Ferrocene Nanosheet Field-Effect Transistors

**DOI:** 10.3390/nano13172469

**Published:** 2023-09-01

**Authors:** Dorra Mahdaoui, Chika Hirata, Kahori Nagaoka, Kun’ichi Miyazawa, Kazuko Fujii, Toshihiro Ando, Manef Abderrabba, Osamu Ito, Shinjiro Yagyu, Yubin Liu, Yoshiyuki Nakajima, Kazuhito Tsukagoshi, Takatsugu Wakahara

**Affiliations:** 1Electronic Functional Macromolecules Group, Research Center for Macromolecules and Biomaterials, National Institute for Materials Science, 1-1 Namiki, Tsukuba 305-0044, Ibaraki, Japan; chikapanman3@gmail.com (C.H.); asagi12@gmail.com (K.N.); fujii.kazuko@nims.go.jp (K.F.); ando.toshihiro@nims.go.jp (T.A.); osamu.ito.b7@tohoku.ac.jp (O.I.); 2Laboratory of Materials, Molecules and Applications, Preparatory Institute for Scientific and Technical Studies, University of Carthage, B.P. 51, La Marsa 2075, Tunisia; mohamedabdelmanef.benabderrabba@ipest.ucar.tn; 3Department of Industrial Chemistry, Faculty of Engineering, Tokyo University of Science, 6-3-1 Niijuku, Katsushika-ku, Tokyo 125-8585, Japan; miyakuni@rs.tus.ac.jp; 4Nano Electronics Device Materials Group, Research Center for Electronic and Optical Materials, 1-1 Namiki, Tsukuba 305-0044, Ibaraki, Japan; yagyu.shinjiro@nims.go.jp; 5RIKEN KEIKI Co., Ltd., 2-7-6, Azusawa, Itabashi-ku, Tokyo 174-8744, Japan; yb-ryu@rikenkeiki.co.jp (Y.L.); y-nakajima@rikenkeiki.co.jp (Y.N.); 6International Center for Materials Nanoarchitectonics (WPI-MANA), National Institute for Materials Science, 1-1 Namiki, Tsukuba 305-0044, Ibaraki, Japan; tsukagoshi.kazuhito@nims.go.jp

**Keywords:** fullerenes, ambipolar to unipolar switch, charge transfer complex, field-effect transistors, nanosheets

## Abstract

Organic cocrystals, which are assembled by noncovalent intermolecular interactions, have garnered intense interest due to their remarkable chemicophysical properties and practical applications. One notable feature, namely, the charge transfer (CT) interactions within the cocrystals, not only facilitates the formation of an ordered supramolecular network but also endows them with desirable semiconductor characteristics. Here, we present the intriguing ambipolar CT properties exhibited by nanosheets composed of single cocrystals of C_70_/ferrocene (C_70_/Fc). When heated to 150 °C, the initially ambipolar monoclinic C_70_/Fc nanosheet-based field-effect transistors (FETs) were transformed into n-type face-centered cubic (fcc) C_70_ nanosheet-based FETs owing to the elimination of Fc. This thermally induced alteration in the crystal structure was accompanied by an irreversible switching of the semiconducting behavior of the device; thus, the device transitions from ambipolar to unipolar. Importantly, the C_70_/Fc nanosheet-based FETs were also found to be much more thermally stable than the previously reported C_60_/Fc nanosheet-based FETs. Furthermore, we conducted visible/near-infrared diffuse reflectance and photoemission yield spectroscopies to investigate the crucial role played by Fc in modulating the CT characteristics. This study provides valuable insights into the overall functionality of these nanosheet structures.

## 1. Introduction

In recent times, low-cost and flexible organic cocrystals have become a subject of considerable research interest. The fascination with these materials originates from their unique chemo-physical properties that set them apart as promising materials for a wide range of applications. Notably, these cocrystals possess high electrical conductivity, making them efficient conductors of electricity, and they also exhibit impressive photoconductivity, enabling them to respond to light and convert it into an electrical signal. Additionally, their photovoltaic properties make them promising candidates for use in advanced solar cell technologies. Furthermore, the tunable luminescent features of these materials offer opportunities for the development of innovative optoelectronic devices.

Organic cocrystals also display superconductivity, allowing them to conduct electricity without any resistance at extremely low temperatures, and thus endowing them with significant potential for cutting-edge electronic and energy applications. Moreover, their ability to efficiently transport both positive and negative charge carriers, known as ambipolar charge carrier transport, makes them versatile components for electronic devices. In fact, achieving concurrent movement of electrons and holes, which is termed ambipolar charge transport, remains a profoundly desirable trait. Ambipolar charge transport holds notable promise, offering the potential to enhance the design of electronic circuits with superior performance, while also showcasing the capabilities of multifunctional organic devices such as transistors that are effective for both light emission and light sensing [1].

Given their low cost and flexibility, organic cocrystals are more advantageous for practical use than the traditional inorganic semiconductors and, in particular, they are suitable for large-scale device production using scalable and affordable methods. This has motivated an active research effort to explore the applications of organic cocrystals in various fields ranging from electronics and energy harvesting to sensors and medical technologies. The ongoing investigations into such organic cocrystals promise to drive progress in materials science and open up new possibilities for the development of novel and efficient technologies [2,3,4,5,6,7,8,9,10,11,12,13].

Organic cocrystals are constructed from two or more different species through noncovalent intermolecular interactions, such as π–π interactions, halogen and hydrogen bonds, and charge transfer (CT) interactions [14], of which the CT interactions are the most prominent. CT interactions occur through charge transfer between the highest occupied molecular orbital (HOMO) of the electron-rich donor and the lowest unoccupied molecular orbital (LUMO) of the electron-deficient acceptor [15,16,17,18]. These interactions are facilitated when strong charge acceptors, such as fullerenes, are paired with suitable charge donors, such as porphyrins and ferrocene (Fc).

Over the years, intensive research has been conducted on CT complexes aimed at formulating the principles for guiding the design of materials with superior characteristics such as high mobility [19] and superconductivity [20]. Recently, Chen et al. successfully developed two-dimensional (2D) C_60_ microsheets into intricately organized nanorod arrays via the CT interactions between rubrene and C_60_, with rubrene serving as the structure-directing agent [21].

Recently, significant attention has been paid to developing technologically significant applications for CT complexes, such as thermoelectric devices [22,23], photoconductors [24], sensors [25], ferroelectrics [26], and organic field-effect transistors (OFETs) [27,28,29], in which CT complexes can function as either organic metals or organic semiconductors. Ambipolar transport, wherein positive and negative charge carriers can both be transported concurrently within the same semiconducting channel, can broaden the scope of semiconductor applications. Achieving ambipolar charge transport is often challenging when working with individual semiconductor components. By co-assembling p- and n-type semiconductors, we can successfully overcome this limitation [30]. This form of transport has been observed in several donor/acceptor (D–A) cocrystals [31,32]. Hence, fabricating organic cocrystals provides a feasible means of obtaining ambipolar materials; nevertheless, the preparation of these semiconductors remains a challenge owing to complex organic syntheses. However, using a simple and efficient liquid–liquid interfacial precipitation method for the synthesis of low-dimensional nanomaterials based on fullerenes [33,34,35,36,37], our group has successfully synthesized cocrystals of fullerenes with ambipolar transport characteristics [1,38,39]. Recently, we successfully demonstrated the capability to irreversibly switch the electrical properties of C_60_/Fc nanosheets from ambipolar to unipolar by thermal stimulation [39]. However, the C_60_/Fc nanosheet devices were thermally unstable—more than half of the C_60_/Fc nanosheet devices lost their ambipolar properties below 80 °C [39]. In a related work, in 2014, Osonoe et al. published a study on monoclinic C_70_/Fc cocrystal nanosheets fabricated with a size of roughly 4.6 μm and featuring (Fc)_2_-C_70_ motifs [40]. When heated, the monoclinic C_70_/Fc nanosheets were transformed to fcc C_70_ nanosheets by losing Fc. Osonoe et al. also reported that this structural change due to the evaporation of Fc in the C_70_/Fc nanosheets rarely occurred at temperatures lower than 150 °C. By contrast, such a change easily occurred in the case of the C_60_/Fc nanosheets, as reported by our group [39]. This implies that compared to C_60_/Fc nanosheets, C_70_/Fc nanosheets could be a superior candidate material for thermally stable devices.

Here, we present the fabrication of ambipolar C_70_/Fc cocrystal nanosheet FETs, elucidate the effect of the temperature on the crystal composition, and relate the electrical and structural properties of the nanosheets. In fact, some organic semiconductors exhibit the remarkable capability of modulating their electrical properties in response to external stimuli such as temperature variations, exposure to light, mechanical forces, and interactions with chemical or biological agents [39]. In this study, we have opted to examine the impact of heat treatment on the electrical transport properties of these materials due to its straightforwardness, ease of implementation, and the absence of the need for expensive experimental equipment.

Additionally, we examine the role played by Fc in the electronic properties of the nanosheets using photoemission yield spectroscopy in air (PYSA). Moreover, we explore the key role of Fc in the switching of the ambipolar C_70_/Fc nanosheets to n-type fullerene nanosheets caused by the sublimation of Fc upon the heating of the nanosheets to 150 °C.

## 2. Materials and Methods

C_70_/Fc nanosheets were synthesized following a previously reported method [40]. A visible/near-infrared diffuse reflectance spectrometer (V-570, JASCO, Tokyo, Japan) equipped with an integrating sphere was used to study their optical properties. Moreover, PYSA measurements were conducted using a photoelectron spectrometer (AC-3, RIKEN KEIKI Co., Ltd., Tokyo, Japan) equipped with a monochromated D2 lamp. C_70_/Fc nanosheet FETs were fabricated using a methodology outlined in previous studies [39]. The electrical transport properties of the fabricated FETs were evaluated in a controlled-environment glove box using a semiconductor analyzer (B2902A and E5272A, Agilent, Santa Clara, CA, USA).

## 3. Results and Discussion

The synthesis procedure for the C_70_/Fc nanosheets is comprehensively described in the Appendix A. The nanosheets formed at the interface between toluene and isopropyl alcohol (IPA) possessed hexagonal morphology with long and short axes, as reported by Osonoe et al. [40]. To study the CT properties of the C_70_/Fc nanosheets, we prepared bottom-gate, bottom-contact FETs using these sheets. This entailed depositing a solution containing C70/Fc nanosheets onto a substrate with prepatterned gold source–drain electrodes using drop casting. The electrodes had a channel width of 10,000 μm and their length ranged from 2 to 10 μm. The gate dielectric was composed of SiO_2_ with a thickness of 300 nm. Before the deposition, the gold electrodes underwent treatment with self-assembled monolayers of undecanethiol, whereas the SiO_2_ interface became hydrophobic through treatment with hexamethyldisilazane [41].

Figure 1 shows the transfer characteristics (I_D_ vs.V_G_) of the FET. The measurements were performed at room temperature in an N_2_ environment to ensure darkness. Notably, the transfer curves exhibit a distinctive V-shape, where one arm corresponds to the electron transport (n-type), whereas the other arm indicates the hole transport (p-type). The calculated electron and hole mobilities in the C_70_/Fc cocrystals presented in Figure 1a are approximately 10^−5^ and 10^−7^ cm^2^ V^−1^ s^−1^, respectively, which is slightly lower compared to the transport properties of the C_60_/Fc nanosheets previously reported by us which exhibited the electron and hole mobility values of 10^−3^ and 10^−5^ cm^2^ V^−1^ s^−1^, respectively [39]. The lower symmetry of the C_70_ molecules may explain the larger centroid–centroid distance in the C_70_/Fc cocrystal, resulting in a reduced intermolecular electronic coupling between the fullerene molecules and a subsequent decrease in the electron mobility. Moreover, Goudappagouda et al. achieved successful fabrication of ambipolar D-A single-crystalline assemblies composed of 1D arrangements of 5,10,15,20-tetraphenylporphyrins (H_2_TPP, ZnTPP), and fullerene (C_60_) [42]. These assemblies exhibited superior ambipolar mobility compared to our device. Furthermore, an outstanding cocrystal ambipolar FET device based on triple-decker mixed (phthalocyaninato) (porphyrinato) yttrium(III) and fullerene cocrystals reported remarkably high carrier mobilities of 3.72 and 2.22 cm^2^ V^−1^ s^−1^ for holes and electrons, respectively [27]. However, despite the lower observed mobilities, our results are significant in that they demonstrate the simultaneous transport of electrons and holes (ambipolar charge transport) in C_70_/Fc nanosheets. This feature is highly desirable because it enables the design of improved electronic circuits and bifunctional organic devices, such as light-emitting and light-sensing transistors. It is important to note that very little research has been conducted on self-assembled aggregates with ambipolar transport properties, and that not all self-assembled fullerene–donor complexes exhibit this characteristic [35,43]. Due to the strong potential for improvement, we emphasize the importance of further optimizing the device to achieve higher carrier mobilities. Recent findings by Liu et al. demonstrate the effectiveness of micro/nanoscale interface passivation combined with flexible polymer dielectrics in enhancing the electrical performance of OFETs [44]; this may be an effective approach for the optimization of devices based on C_70_/Fc nanosheets.

The ambipolar characteristics observed in the C_70_/Fc nanosheets were also found in the C_60_/Fc nanosheets [34]. However, this result is in stark contrast to our previous findings, in which C_70_ nanosheet-based FETs displayed n-type behavior only [45]. Additionally, Fc is not typically recognized as a semiconductor material [39]. Considering these findings, we highlighted that the interplay between Fc and the C_70_ molecules plays a key role in determining the charge-transfer characteristics of the C_70_/Fc nanosheets.

In our previous study [39], we observed irreversible switching in the semiconducting characteristics of the device, i.e., the transition from ambipolar to unipolar, which matched the thermally induced structural transition of the C_60_/Fc nanosheets. Additionally, Osonoe et al. reported the conversion of monoclinic C_70_/Fc nanosheets to fcc C_70_ nanosheets under heat treatment, with the hexagonal shape and size of the nanosheets remaining largely unchanged [40]. These discoveries motivated us to examine the impact of the heat treatment on the electrical transport properties of the C_70_/Fc nanosheets and the role played by Fc in this context. Recently, a similar switching device based on a periodically boron-doped (nitrogen-doped) armchair graphene nanoribbon was theoretically proposed by Wang et al. [46].

To eliminate the possibility that the switching behavior of the C_70_/Fc nanosheets was caused by the absorption or desorption of oxygen impurities during annealing, we conducted the FET measurements and annealing sequentially in a glove box, thus ensuring a consistently oxygen-free environment. Consequently, the oxygen content remained constant. Thus, the transfer characteristics of the C_70_/Fc nanosheets were investigated following annealing under N_2_ atmosphere, as illustrated in Figure 1b and Figure 2. As the temperature reached 80 °C, we observed an almost constant electron mobility (~10^−5^ cm^2^ V^−1^ s^−1^), accompanied by a minor reduction in hole mobility (10^−7^–10^−8^ cm^2^ V^−1^ s^−1^). Subsequent measurements conducted under negative V_G_ conditions, after annealing at 150 °C, revealed a reduction in the conductivity of the nanosheets. Such complete elimination of the hole transport clearly demonstrated n-type behavior, indicating a full and irreversible change of the device’s semiconductor characteristics from ambipolar to unipolar. By contrast, previous studies demonstrated that FET devices utilizing pristine C_70_ materials fabricated via solution processes, such as C_70_ nanosheets [46] and C_70_ single-crystal needles [47], did not exhibit a switching of their semiconducting properties when measured in an N_2_ environment. These observations and the results of previous studies lead us to attribute the switching behavior observed in the current study to the ablation of the Fc molecules during the heat treatment. We investigated more than 20 functional ambipolar C_70_/Fc nanosheet devices and observed the cessation of hole transport after annealing at 80 °C (23%) and 150 °C (77%). In our previous study [39], we observed a loss of hole transport in 12.5% and 62.5% of the examined C_60_/Fc nanosheet devices after annealing at 60 and 80 °C, respectively. Furthermore, only 25% of the devices switched their behavior at 150 °C. These results indicate that the C_70_/Fc nanosheet FETs were more thermally stable than those composed of C_60_/Fc nanosheets. The observed electron mobility (~10^−5^ cm^2^ V^−1^ s^−1^) in the C_70_ nanosheets after annealing at 150 °C was nearly identical to that in the C_70_/Fc nanosheet FETs after annealing at 80 °C. We also found an increase in off-state current (I_d_ at V_g_ of 0 V) in Figure 2 after annealing at 150 °C. In order to determine the origin of this increased off-state current, we measured n-type FET transfer curves from −60 to 60 V (Appendix A). From the data in Appendix A, we conclude that the C_70_ nanosheets FETs behave as n-channel, normally-on type FETs. The similar normally-on type FETs were also reported for fullerene nanomaterials-based FETs after annealing [47,48]. This will be due to the doping by residual ferrocene and/or partial polymerization of fullerenes by annealing. Therefore, in order to decrease the off-state current, it is considered effective to optimize the device structure, for example, to reduce the operating voltage by changing from the bottom to the top contact structure and/or by changing the electrode metal from Au to Al. 

Understanding the charge transport properties of C_70_/Fc nanosheets requires detailed information about their electronic structure. In this study, we conducted PYSA measurements on the C_70_/Fc cocrystals, Fc, and C_70_ powder. PYSA is an effective method for exploring the electronic and electrical characteristics of molecular semiconductors [38,49]. In a previous study, we successfully utilized PYSA to evaluate the electronic structure of the C_60_/Fc nanosheets with high sensitivity [39]. The energy level diagrams of the materials can be estimated using the ionization energy (*I_s_*) measured by PYSA and the energy gap (*E_g_*) derived from the ultraviolet-visible absorption spectra of the materials.

The absorption spectrum of the C_70_/Fc cocrystals has not been reported so far. Therefore, we first measured the diffuse reflectance spectra of the C_70_/Fc nanosheets (Figure 3) before and after annealing, as well as that of the C_70_ powder. Notably, the C_70_ powder exhibited broad absorption within the 700–400 nm range, whereas the C_70_/Fc cocrystals demonstrated distinct absorption in the longer-wavelength region (up to 1000 nm). Subtracting the standardized absorption spectrum of the C_70_ powder from that of the C_70_/Fc cocrystals revealed an additional absorption in the 1000–700 nm range, with a peak absorption at 800 nm, which was identified as the CT absorption band of the C_70_/Fc nanosheets. The CT transition energy corresponding to the peak at 800 nm is estimated to be 1.55 eV, which is very close to that of the C_60_/Fc nanosheets reported by our group [34]. Interestingly, upon heating, the 800 nm band gradually disappeared, confirming its nature as a CT band. However, in contrast to that of the C_60_/Fc nanosheets, the CT band of the C_70_/Fc nanosheets did not completely disappear even after heating to 250 °C under vacuum. This result also indicates the fact that the C_70_/Fc nanosheets were more thermally stable than the C_60_/Fc nanosheets.

Figure 4a illustrates the PYSA spectra of the C_70_/Fc nanosheets and C_70_ powder [38]. The *I_s_* of the C_70_/Fc nanosheets was determined to be 5.61 eV, which was 0.6 eV lower than that of the C_70_ powder. Figure 4b presents the energy level diagrams of the C_60_/Fc nanosheets [39], Fc [39], C_70_/Fc nanosheets, and C_70_ powder [38], estimated using the measured *I_s_* and *E_g_
*_values_. The *E_g_* of the C_70_/Fc nanosheets was derived from the UV-visible absorption spectrum obtained in this study. These energy level diagrams reveal that the cocrystallization of C_70_ with Fc improved the p-type FET characteristics due to the reduction of the barrier for the hole injection from the gold electrode to the HOMO of the C_70_/Fc nanosheet, which aligned closely with that of the Fc powder. By contrast, the energy level of the LUMO of the C_70_/Fc nanosheets was close to that of C_70_; this gave rise to the n-type FET characteristics of the C_70_/Fc nanosheets. These findings are consistent with the ambipolar transport properties observed in the C_70_/Fc nanosheets and also with the recent findings for the C_60_/Fc nanosheets [39]. Thus, the obtained results for the HOMO and LUMO can be ascribed to the CT interactions between C_70_ and Fc. These results provide evidence for the significant contribution made by the intermolecular interplay between the C_70_ and Fc molecules within the nanosheets, thus highlighting its crucial role in determining the ambipolar CT properties exhibited by the C_70_/Fc nanosheets.

Our research results provide strong evidence for the exceptional properties of two-dimensional (2D) C_70_/Fc nanosheets, which have attracted significant interest as functional materials. These nanosheets possess a remarkable combination of advantageous characteristics such as a large specific surface area, narrow bandgap, porosity, excellent electron transfer capability, broad light absorption range, and remarkable stability in various environmental conditions. These characteristics are similar to those of three-dimensional (3D) transition metal oxides that also exhibit significant advantages in ultrafast optics [50]. Previous studies have demonstrated the promising potential of photonics devices based on such oxide materials in various applications, such as ultrafast lasers [51,52,53,54,55], high-performance sensors, and fiber-optic communications. These compelling findings have led to exciting opportunities for integrating 3D transition metal oxides with novel fibers, thus advancing the technology of ultrafast optics devices [50]. Therefore, our study suggests that C_70_/Fc nanosheets, combined with a versatile platform such as dual-core, or pair-hole fiber (DCPHF), have considerable promise as candidates for driving innovations in ultrafast photonics.

Furthermore, organic materials [56], transition metal oxides nanomaterials, and metal-organic framework (MOF) materials have gained considerable attention in the field of ultrafast photonics and nonlinear optical devices. These materials share some interesting properties with C_70_/Fc nanosheets, such as high optical absorption and a relatively narrow bandgap, making them highly intriguing for advanced photonics and nonlinear optical applications.

For instance, Li et al. successfully employed porous dodecahedron rGO-Co_3_O_4_ as an outstanding nonlinear optical modulator by calcining an MOF template [57]. Additionally, Zhang et al. demonstrated the potential of NiO-MOF to achieve harmonic mode-locking at an fs level of more than 400 MHz, supporting the use of MOF materials in advanced photonics [58]. Similarly, porous MOF-derived CuO octahedral oxide has been utilized as a saturable absorber (SA) in fiber lasers [59].

As reported by Zhang et al., nanoscale PbTe has shown promise as a material for ultrafast photonic applications and has driven advances in the development of semiconductor crystal-based optical devices [60]. Furthermore, research by Zhao et al. has laid the groundwork for advanced photonics based on Cu_2_O nanocube materials [61].

Considering the exceptional properties of the C_70_/Fc nanosheets, the investigation of these nanomaterials can pave the way for advanced photonics based on fullerene (D-A) cocrystals. The synthesis and characterization of such materials are of utmost importance because ultrafast photonics is a dynamic and rapidly evolving research area that is continuously pushing the boundaries of the science and technology of light-matter interactions.

Indeed, our findings provide an exciting opportunity for the further exploration and development of these materials in ultrafast photonics applications, with the potential to revolutionize various industries, ranging from telecommunications and data processing to cutting-edge research in fundamental physics and quantum technology. Ongoing research and advances in this direction are crucial for unlocking the full potential of ultrafast photonics and shaping the future of light-based technologies.

## 4. Conclusions

We achieved successful fabrication of ambipolar FETs utilizing C_70_/Fc nanosheets, which displayed enhanced thermal stability compared to FETs based on C_60_/Fc nanosheets. This improvement was realized through a straightforward cocrystallization process. Remarkably, despite initially containing only n-type semiconductors, the devices exhibited ambipolar charge transport characteristics. However, through annealing at 150 °C in an inert atmosphere and eliminating Fc, we achieved a transformation to stable n-type C_70_ nanosheet-based FETs, accompanied by irreversible changes in the device’s semiconducting properties. Notably, the higher stability of the C_70_/Fc nanosheet devices is attributed to the strong interaction between C_70_ and Fc. Additionally, our findings revealed a CT band with peak absorption at 800 nm in the diffuse reflectance spectrum of the C_70_/Fc nanosheets that gradually disappeared upon heating. Energy level diagrams derived from the PYSA measurements and absorption spectra showed that cocrystallization of C_70_ with Fc significantly improved the p-type FET characteristics by reducing the barrier for the hole injection from the gold electrode to the highest occupied molecular orbital (HOMO) of the C_70_/Fc nanosheet.

In summary, our study underscores the importance and necessity of developing novel synthesis methods for nanomaterials, because our findings shed light on the complex electronic properties of these materials. Our work highlights the potential of single cocrystals containing appropriate donor–acceptor molecules to exhibit ambipolar behavior, even in the absence of p-type semiconductors. These results pave the way for exciting future research in the field of nanomaterial synthesis and electronics.

## Figures and Tables

**Figure 1 nanomaterials-13-02469-f001:**
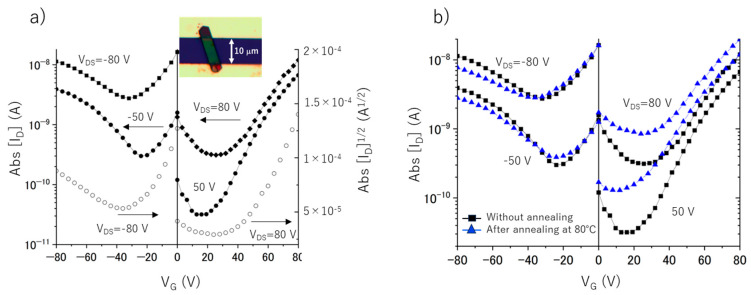
Transfer characteristics (drain current (I_D_) vs. gate voltage (V_G_)) of C_70_/Fc nanosheets in the dark for positive and negative gate biases (**a**) without annealing, and (**b**) after annealing at 80 °C. The solid lines show I_D_ vs. V_G_ for various drain–source voltages (V_DS_), and the open symbols show the square root of I_D_ (right vertical axis). Inset: optical microscopy image of the typical FET based on a hexagonal C_70_/Fc nanosheet with long and short axes.

**Figure 2 nanomaterials-13-02469-f002:**
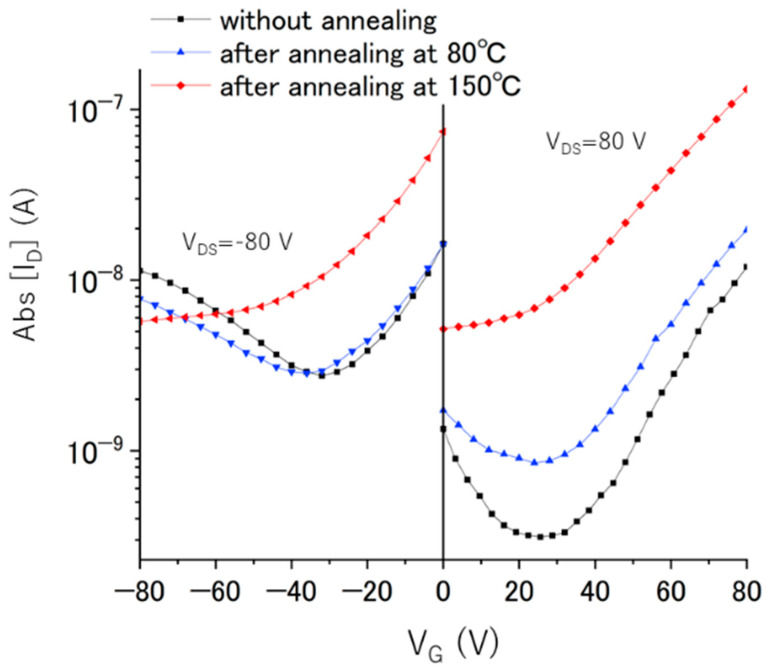
Transfer characteristics (I_D_ vs. V_G_) of the C_70_/Fc nanosheets in the dark after annealing.

**Figure 3 nanomaterials-13-02469-f003:**
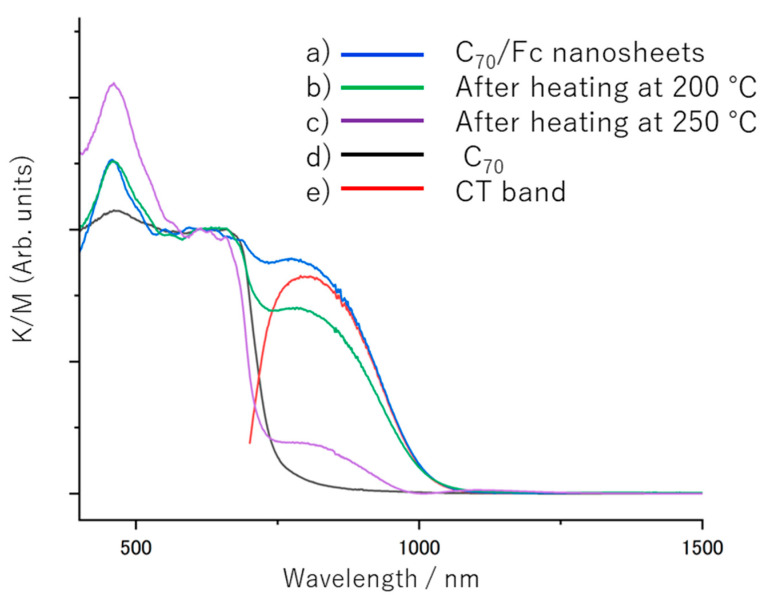
Diffuse reflectance spectra in the visible and near-infrared regions (K/M stands for Kubelka–Munk function, which is a measure of the absorbance): (a) C_70_/Fc nanosheets at room temperature; (b) C_70_/Fc nanosheets after heating to 200 °C; (c) C_70_/Fc nanosheets after heating to 250 °C; (d) C_70_ powder; (e) subtraction of the normalized absorption spectra at 610 nm, i.e., (a)–(d).

**Figure 4 nanomaterials-13-02469-f004:**
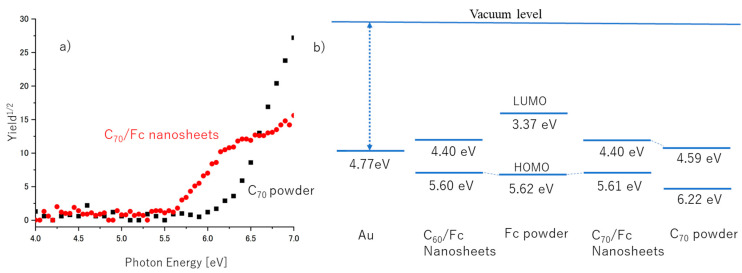
(**a**) PYSA spectra of C_70_/Fc nanosheets and C_70_ powder; (**b**) energy level diagrams of C_60_/Fc nanosheets, Fc powder, C_70_/Fc nanosheets, and C_70_ powder.

## Data Availability

The data presented in this study are available on reasonable request from the corresponding author.

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
