# Peer review of "Ambipolar to Unipolar Conversion in C_70_/Ferrocene Nanosheet Field-Effect Transistors"

_nanomaterials, 2023, doi:10.3390/nano13172469_

Round 1

Reviewer 1 Report

Mahdaoui et al. fabricated C70/Fc nanosheet-based FETs through a cocrystallization process. Furthermore, they discovered that by annealing the devices at 150 °C in an inert atmosphere, the ambipolar C70/Fc nanosheet-based FETs could be transformed into n-type C70 nanosheet-based FETs by eliminating Fc. This thermal treatment induced structural modifications in the cocrystal and resulted in an irreversible switchover of the device's semiconducting properties from ambipolar to unipolar. Overall, the idea received my attention and the methodology is technically sound. However, there are some specific issues the authors should address by making modifications before we can proceed and positive action can be taken.

  1. The conclusions section reads rather long and should be streamlined.

  2. Scientific notation should be set as 1×10 with a superscript −7 as opposed to the computer programming style of 1E-7.

  3. Please increase the font size in figures, in order to make sure that all micrographs have scale markers, and that all scale markers and text in figures are large enough to be easily read.

  4. Compare the properties of your material to the state-of-the-art in literature and industry.

  5. The authors mainly investigated the charge transport characteristics, i.e., switching behaviors. Have the authors noticed some studies related to this point? i.e., [Switching behavior of a heterostructure based on periodically doped graphene nanoribbon. Phys. Rev. Appl. 2021, 16, 024030, doi:10.1103/PhysRevApplied.16.024030]…

Author Response

Ms. Ref. No.:  nanomaterials-2532676

Title: Ambipolar to unipolar conversion in C70/Ferrocene nanosheet field-effect transistors

Response to the Reviewers’ comments

Dear Reviewers,

We are very grateful for your constructive comments to our manuscript. The manuscript was revised in accordance with all your comments and queries, which are valuable in improving the quality of our manuscript. All changes made to the manuscript are marked in red. We hope that the new version of our manuscript meets the high standards for publication in Nanomaterials.

Sincerely, on behalf of all authors

Dr. Dorra Mahdaoui

Laboratory of Materials, Molecules and Applications,

Preparatory Institute for Scientific and Technical Studies,Tunisia.

Phone number: 00216 52 30 91 45

Email address: dorra.mahdaoui@ipest.ucar.tn

Reviewers ‘comments:

Reviewer #1: Mahdaoui et al. fabricated C70/Fc nanosheet-based FETs through a cocrystallization process. Furthermore, they discovered that by annealing the devices at 150 °C in an inert atmosphere, the ambipolar C70/Fc nanosheet-based FETs could be transformed into n-type C70 nanosheet-based FETs by eliminating Fc. This thermal treatment induced structural modifications in the cocrystal and resulted in an irreversible switchover of the device's semiconducting properties from ambipolar to unipolar. Overall, the idea received my attention and the methodology is technically sound. However, there are some specific issues the authors should address by making modifications before we can proceed and positive action can be taken.

  1. The conclusions section reads rather long and should be streamlined.

Answer: Thank you for your valuable feedback; we have considered your suggestions in revising the conclusion.

Revised Conclusion:

We achieved the successful fabrication of ambipolar FETs utilizing C70/Fc nanosheets, which displayed enhanced thermal stability compared to FETs based on C60/Fc nanosheets. This improvement was realized through a straightforward co-crystallization process. Remarkably, despite initially containing only n-type semiconductors, the devices exhibited ambipolar charge transport characteristics. However, through annealing at 150°C in an inert atmosphere and eliminating Fc, we achieved a transformation to stable n-type C70 nanosheet-based FETs, accompanied by irreversible changes in the device's semiconducting properties. Notably, the higher stability of C70/Fc nanosheet devices is attributed to the strong interaction between C70 and Fc.  Additionally, our findings demonstrated a CT band with peak absorption at 800 nm in the diffuse reflectance spectrum of the C70/Fc nanosheets, which gradually disappeared upon heating. Energy level diagrams derived from PYSA measurements and absorption spectra showed that cocrystallization of C70 with Fc significantly improved the p-type FET characteristics by reducing the hole injection barrier between the gold electrode and the highest occupied molecular orbital (HOMO) of the C70/Fc nanosheet

In summary, this study represents a crucial advancement in the synthesis of nanomaterials with intriguing electronic properties. It highlights the potential of single cocrystals containing appropriate donor–acceptor molecules to exhibit ambipolar behavior, even in the absence of p-type semiconductors. These results pave the way for exciting future research in the field of nanomaterial synthesis and electronic applications

  1. Scientific notation should be set as 1×10 with a superscript −7 as opposed to the computer programming style of 1E-7.

Answer: Thanks for the comment; we update the scientific notation in the revised manuscript.

  1. Please increase the font size in figures, in order to make sure that all micrographs have scale markers, and that all scale markers and text in figures are large enough to be easily read.

Answer: Thank you very much for pointing this out. We have carefully revised Figures

  1. Compare the properties of your material to the state-of-the-art in literature and industry.

Answer: Thank you for your valuable suggestions. We have incorporated the new references and included a discussion comparing the properties of our material with high-performance counterparts.

  1. «The calculated electron and hole mobilities in the C70/Fc cocrystals, as presented in Figure S1, are slightly lower, approximately 10–5and 10–7 cm2 V−1 s−1, respectively, compared to our previously reported transport properties of C60/Fc nanosheets, which exhibited electron and hole mobility values of 10-3 and 10-5 cm2 V−1 s−1, respectively [39]. The lower symmetry of C70 compared to C60 molecules may explain the larger centroid–centroid distance in the C70/Fc cocrystal, resulting in reduced intermolecular electronic coupling between fullerene molecules and a subsequent decrease in electron mobility. Moreover, Goudappagouda et al. achieved successful fabrication of ambipolar donor-acceptor single-crystalline assemblies, comprising 1D arrangements of 5,10,15,20-tetraphenylporphyrins (H2TPP, ZnTPP), and fullerene (C60)[42]. These assemblies exhibited superior ambipolar mobility in comparison to our device. In contrast, an outstanding cocrystal ambipolar OFET device based on triple-decker mixed (phthalocyaninato) (porphyrinato) yttrium(III) and fullerene cocrystals reported remarkably high carrier mobilities of 3.72 and 2.22 cm2 V−1 s−1 for holes and electrons, respectively[43], highlighting the relatively lower performance of our results. However, despite the lower mobilities observed, the significance lies in the simultaneous transport of electrons and holes (ambipolar charge transport) in C70/Fc nanosheets. This feature is highly desirable as it enables the design of improved electronic circuits and bifunctional organic devices, such as light-emitting and light-sensing transistors. It is worth noting that very little research has been conducted on self-assembled aggregates with ambipolar transport properties, and not all self-assembled fullerene–donor complexes exhibit this characteristic [35, 44]. With strong belief in the potential for improvement, we emphasize the importance of optimizing the device further to achieve higher carrier mobilities. Recent findings by Liu et al. demonstrate the effectiveness of micro/nanoscale interface passivation combined with flexible polymer dielectrics in enhancing the electrical performance of OFETs, which can be a solution.»

We added also this parts to the revised manuscript.

  1. «The CT transitionenergy corresponding to the maximum at 800 nm is estimated as 1.55 eV which is very close to that of C60/Fc nanosheets reported by our group [34].»

References (Newly added)

[42] Goudappagouda, Gedda; M., Kulkarni; G. U.; Babu, S. S. One-Dimensional Porphyrin–Fullerene (C60) Assemblies: Role of Central Metal Ion in Enhancing Ambipolar Mobility. Chemistry - Eur. J. 2018, 24(30), 7695–7701. DOI:10.1002/chem.201800197

[43] Liu, Z.; Ju, Z.; Ma, S.; Li, W.; Chen, J.; Yang, B.; Zhang, J. Organic charge‐transfer complex based microstructure interfaces for solution‐processable organic thin‐film transistors toward Multifunctional Sensing. Adv. Electron. Mater. 2023, DOI:10.1002/aelm.202300205

[44] Zhang, J.; Tan, J.; Ma, Z.; Xu, W.; Zhao, G.; Geng, H.; Di, C.; Hu, W.; Shuai, Z.; Singh, K.; Zhu, D.  Fullerene/sulfur-bridged annulene cocrystals: Two-dimensional segregated heterojunctions with ambipolar transport properties and photoresponsivity. Journal of J. Am. Chem. Soc. 2013, 135(2), 558–561. DOI:10.1021/ja310098k

  1. The authors mainly investigated the charge transport characteristics, i.e., switching behaviors. Have the authors noticed some studies related to this point? i.e., [Switching behavior of a heterostructure based on periodically doped graphene nanoribbon. Phys. Rev. Appl. 2021, 16, 024030, doi:10.1103/PhysRevApplied.16.024030]…

Answer: Thank you very much for your important suggestions. We added the new reference.

«Recently, a similar switching device based on a periodically boron-doped (nitrogen-doped) armchair graphene nanoribbon was theoretically proposed by Wang et al [47]. »

Reference (Newly added)

[47] Wang, S.; Hung, N. T.; Tian, H.; Islam, M. S.; Saito, R. Switching behavior of a heterostructure based on periodically doped graphene nanoribbon. Physical Review Applied. 2021, 16, 024030. DOI:10.1103/physrevapplied.16.024030.

We tried our best to improve the manuscript and made some changes in the manuscript. We appreciate for Editors/Reviewers’ warm work earnestly, and hope that the correction will meet with approval.

Once again, thank you very much for your comments and suggestions.

Reviewer 2 Report

This paper present the intriguing ambipolar charge transfer (CT) properties exhibited by nanosheets composed of single cocrystals of C70/ferrocene (C70/Fc). When subjected to a temperature of 150 °C, the initially ambipolar monoclinic C70/Fc nanosheet-based field-effect transistors (FETs) were transformed into n-type face-centered cubic (fcc) C70 nanosheet-based FETs owing to the elimination of Fc. This thermally-induced alteration in the crystal structure was accompanied by an irreversible switch-over in the semiconducting behavior of the device; thus, the device transitions from ambipolar to unipolar. Furthermore, conducted visible/near-infrared diffuse reflectance and photoemission yield spectroscopies to investigate the crucial role played by Fc in modulating the CT characteristics. In order to improve this work, the following questions can be considered and the manuscript need to be revised accordingly.

(1)The authors are suggested to compare and discussion some recent works in the ultrafast photonics fields. (i.e. Ultrafast Science, 2022, 9767251, 16, 2022; Ultrafast Science, 2022, 9870325, 6, 2022; Ultrafast Science, 3, 0006, 2023; Phys. Rev. Lett., 121, 023905 2018; Laser Photon. Rev. 13, 1800333, 2019; Phys. Rev. Lett., 123, 093901, 2019).

(2) In this paper,describe the relationship between electricity and structure.May I ask whether this step is realized ? Could you give us more details?

(3) In this paper, more than 20 kinds of bipolar C70/Fc nano-devices are studied, and the hole transport stops after 80 °C (23%) and 150 °C(77%) retreats. It is suggested to list the information of nano-devices in order to be more convincing.

(4) It is suggested to put the S1 diagram in the article, so as to watch the comparison results.

(5) This paper mentions the effect of heat treatment on the electrical transport properties of C70/Fc nanosheets and the role of Fc in this context. Why use heat treatment? What are the advantages?

 (6) The manuscript should mention some related work that is discussed in more depth. Namely, Applied Materials and Interfaces, 9(5), 2101933, (2022); Optics and Laser Technology, 146, 107546, (2022); Journal of Materials Chemistry C, 8, 14386-1439, (2020); Applied Materials Today Inc. 28, 101546, (2022); ACS Photonics, 7, 9, 2440-2447. (2020; Optics and Laser Technology, 151, 108016, (2022)).

Author Response

Ms. Ref. No.:  nanomaterials-2532676

Title: Ambipolar to unipolar conversion in C70/Ferrocene nanosheet field-effect transistors

Response to the Reviewers’ comments

Dear Reviewers,

We are very grateful for your constructive comments to our manuscript. The manuscript was revised in accordance with all your comments and queries, which are valuable in improving the quality of our manuscript. All changes made to the manuscript are marked in red. We hope that the new version of our manuscript meets the high standards for publication in Nanomaterials.

Sincerely, on behalf of all authors

Dr. Dorra Mahdaoui

Laboratory of Materials, Molecules and Applications,

Preparatory Institute for Scientific and Technical Studies,Tunisia.

Phone number: 00216 52 30 91 45

Email address: dorra.mahdaoui@ipest.ucar.tn

Reviewers ‘comments:

Reviewer #2: This paper present the intriguing ambipolar charge transfer (CT) properties exhibited by nanosheets composed of single cocrystals of C70/ferrocene (C70/Fc). When subjected to a temperature of 150 °C, the initially ambipolar monoclinic C70/Fc nanosheet-based field-effect transistors (FETs) were transformed into n-type face-centered cubic (fcc) C70 nanosheet-based FETs owing to the elimination of Fc. This thermally-induced alteration in the crystal structure was accompanied by an irreversible switch-over in the semiconducting behavior of the device; thus, the device transitions from ambipolar to unipolar. Furthermore, conducted visible/near-infrared diffuse reflectance and photoemission yield spectroscopies to investigate the crucial role played by Fc in modulating the CT characteristics. In order to improve this work, the following questions can be considered and the manuscript need to be revised accordingly.

  1. The authors are suggested to compare and discussion some recent works in the ultrafast photonics fields. (i.e. Ultrafast Science, 2022, 9767251, 16, 2022; Ultrafast Science, 2022, 9870325, 6, 2022; Ultrafast Science, 3, 0006, 2023; Phys. Rev. Lett., 121, 023905 2018; Laser Photon. Rev. 13, 1800333, 2019; Phys. Rev. Lett., 123, 093901, 2019).

Answer: Thank you very much for your important suggestions.The following sentences were added in to main text.

«Our research results strongly support the exceptional properties of two-dimensional (2D) C70/Fc nanosheets, which have attracted significant interest as functional materials. These nanosheets possess a remarkable combination of advantageous characteristics, including a large specific surface area, a narrow bandgap, porosity, excellent electron transfer capability, broad light absorption range, and remarkable stability in various environmental conditions. Additionally, three-dimensional (3D) transition metal oxides also exhibit similar properties and significant advantages in ultrafast optics [49].

Previous studies have demonstrated the promising potential of photonics devices based on such materials in various applications, such as ultrafast lasers[50-54], high-performance sensors, and fiber-optic communications. These compelling findings have led to exciting opportunities for integrating  3D transition metal oxides  with novel fibers, thus advancing the technology of ultrafast optics devices[49].

Therefore, our study suggests that C70/Fc nanosheets, combined with a versatile platform like dual-core, pair-hole fiber (DCPHF), hold considerable promise as candidates for driving innovations in ultrafast photonics.»

References (Newly added)

[49] Li, X.; Huang, X.; Han, Y.; Chen, E.; Guo, P.; Zhang, W.; An, M.; Pan, Z.; Xu, Q.; Guo, X.; Huang, X.; Wang, Y.; Zhao, W. High-performance γ-MnO2 dual-core, pair-hole fiber for ultrafast photonics. Ultrafast Sci. 2023, 3, DOI:10.34133/ultrafastscience.0006 .

[50] Guan, M.; Chen, D.; Hu, S.; Zhao, H.; You, P.; Meng, S.  Theoretical insights into ultrafast dynamics in Quantum Materials. Ultrafast Sci. 2022, 2022, 9767251. DOI:10.34133/2022/9767251.

[51] Zhang, Z.; Zhang, J.; Chen, Y.; Xia, T.; Wang, L.; Han, B.; He, F.; Sheng, Z.; Zhang, J. Bessel terahertz pulses from superluminal laser plasma filaments. Ultrafast Sci. 2022, 2022, 9870325.  DOI:10.34133/2022/9870325.

[52] Liu, X.; Yao, X.; Cui, Y. Real-time observation of the buildup of soliton molecules. Phys. Rev. Lett.. 2018, 121, 023905.  DOI:10.1103/physrevlett.121.023905. 

[53] Liu, X.; Pang, M. Revealing the buildup dynamics of Harmonic Mode‐locking states in Ultrafast Lasers.  Laser Photonics Rev. 2019, 13, 1800333. DOI:10.1002/lpor.201800333. 

[54] Liu, X.; Popa, D.; Akhmediev, N.  Revealing the Transition Dynamics from Q Switching to Mode Locking in a Soliton Laser. Phys. Rev. Lett 2019, 123, 093901. DOI:10.1103/physrevlett.123.093901. 

  1. In this paper,describe the relationship between electricity and structure.May I ask whether this step is realized ? Could you give us more details?

Answer: Thanks for the comment. We are sorry for not explaining very well.

Indeed, we have not conducted a specific study focused solely on the correlation between electricity and structure. Instead, we use the term "structure" to encompass composition, morphology, and dimensions. The selection of the synthesis method and solvents, along with their proportions, plays a crucial role in determining the structure of the resulting self-assemblies, thus influencing the electronic properties of these materials. In a previous study, we demonstrated that an increase in the thickness of the synthesized self-assemblies is unfavorable for exciton formation and dissociation during device operation, resulting in a decrease in mobility[35]. Additionally, the symmetry and distances between the different molecules comprising these materials significantly impact the intermolecular coupling between the fullerenes, potentially leading to a negative influence on electron mobility.

The following sentences was added to the main text:

 «The calculated electron and hole mobilities in the C70/Fc cocrystals, as presented in Figure S1, are slightly lower, approximately 10–5 and 10–7 cm2 V−1 s−1, respectively, compared to our previously reported transport properties of C60/Fc nanosheets, which exhibited electron and hole mobility values of 10-3 and 10-5 cm2 V−1 s−1, respectively [39]. The lower symmetry of C70 compared to C60 molecules may explain the larger centroid–centroid distance in the C70/Fc cocrystal, resulting in reduced intermolecular electronic coupling between fullerene molecules and a subsequent decrease  in electron mobility.»

  1. In this paper, more than 20 kinds of bipolar C70/Fc nano-devices are studied, and the hole transport stops after 80 °C (23%) and 150 °C(77%) retreats. It is suggested to list the information of nano-devices in order to be more convincing.

Answer: Thank you very much for pointing this out. We have added information of nano-devices and the list is shown in the reviewers only SI.

  1. It is suggested to put the S1 diagram in the article, so as to watch the comparison results.

Answer: we have made changes as suggested by the reviewer.

5. This paper mentions the effect of heat treatment on the electrical transport properties of C70/Fc nano sheetsand the role of Fc in this context. Why use heat treatment? What are the advantages?

Answer: Thanks for the comment; we are sorry for not explaining this step very well. The following sentences were added to main text.

«Here, we present the fabrication of ambipolar C70/Fc cocrystal nanosheet FETs, elucidate the effect of temperature on the crystal composition, and correlate between their electrical and structural properties. Furthermore, Some organic semiconductors exhibit the remarkable capability of modulating their electrical properties in response to external stimuli, including variations in temperature, exposure to light, mechanical forces, and interactions with chemical or biological agents[39]. In this study, we have opted to investigate the impact of heat treatment on the electrical transport properties of these materials due to its straightforwardness, ease of implementation, and the absence of the need for expensive experimental equipment. »

6. The manuscript should mention some related work that is discussed in more depth. Namely, Applied Materials and Interfaces, 9(5), 2101933, (2022); Optics and Laser Technology, 146, 107546, (2022); Journal of Materials Chemistry C, 8, 14386-1439, (2020); Applied Materials Today Inc. 28, 101546, (2022); ACS Photonics, 7, 9, 2440-2447. (2020; Optics and Laser Technology, 151, 108016, (2022)).

Answer: Thanks for the insightful comment. The following sentences were added to main text.

«Furthermore, organic materials[55], transition metal oxides nanomaterials, and metal-organic framework (MOF) materials have gained significant attention in the field of ultrafast photonics and nonlinear optical devices. they share some interesting properties with C70/Fc nanosheets. These properties include large optical absorption and a relatively narrow bandgap, making them highly intriguing for advanced photonics and non-linear optical applications.

For instance, Li et al. successfully employed porous dodecahedron rGO-Co3O4 as an outstanding nonlinear optical modulator by calcining the MOF template [56]. Additionally, Zhang et al. demonstrated the potential of NiO-MOF in achieving harmonic mode-locking at the fs level of more than 400 MHz, supporting the use of MOF materials in advanced photonics [57]. Similarly, porous MOF-derived CuO octahedral has been utilized as a saturable absorber (SA) in fiber lasers[58].

Nano PbTe, as reported by Zhan et al., has shown promise as a material for ultrafast photonic applications and has driven advancements in semiconductor crystal-based optical devices [59]. Furthermore, Zhao et al.'s research has laid the groundwork for advanced photonics based on Cu2O nanocube materials [60].

Considering the exceptional properties of C70/Fc nanosheets, the investigation of these nanomaterials could pave the way for advanced photonics centered around fullerene (D-A) cocrystals. The synthesis and characterization of such materials are of utmost importance as ultrafast photonics represents a dynamic and rapidly evolving research area, continuously pushing the boundaries of light-matter interactions.

Indeed, our findings provide an exciting opportunity for further exploration and development of these materials in ultrafast photonics applications, with the potential to revolutionize various industries, ranging from telecommunications and data processing to cutting-edge research in fundamental physics and quantum technology. Ongoing research and advancements in this direction are crucial for unlocking the full potential of ultrafast photonics and shaping the future of light-based technologies.»

References (Newly added)

[55] Li, X.; Xu, W.; Wang, Y.; Zhang, X.; Hui, Z.; Zhang, H.; Wageh, S.; Al-Hartomy, O. A.; Al-Sehemi, A. G. Optical-intensity modulators with PBTE thermoelectric nanopowders for ultrafast photonics. Appl. Mater. Today. 2022, 28, 101546. DOI:10.1016/j.apmt.2022.101546

[56] Li, X.; An, M.; Li, G.; Han, Y.; Guo, P.; Chen, E.; Hu, J.; Song, Z.; Lu, H.; Lu, J.  Mof‐derived porous dodecahedron rgo‐co            3            O            4            for robust pulse generation. Adv. Mater. Interfaces. 2022, 9, 2101933. DOI:10.1002/admi.202101933

[57] Zhang, C.; Liu, J.; Gao, Y.; Li, X.; Lu, H.; Wang, Y.; Feng, J.; Lu, J.; Ma, K.; Chen, X.  Porous nickel oxide micron polyhedral particles for high-performance ultrafast photonics. Opt. Laser Technol. 2022, 146, 107546. DOI:10.1016/j.optlastec.2021.107546

[58] Zhao, Y.; Wang, W.; Li, X.; Lu, H.; Shi, Z.; Wang, Y.; Zhang, C.; Hu, J.; Shan, G. Functional porous MOF-derived cuo octahedra for harmonic Soliton Molecule Pulses generation. ACS Photonics. 2020, 7, 2440–2447. DOI:10.1021/acsphotonics.0c00520

[59] Zhang, C.; Li, X.; Chen, E.; Liu, H.; Shum, P. P.; Chen, X. Hydrazone Organics with third-order nonlinear optical effect for femtosecond pulse generation and control in the L-band. Opt. Laser Technol. 2022, 151, 108016. DOI:10.1016/j.optlastec.2022.108016

[60] Li, X.; Feng, J.; Mao, W.; Yin, F.; Jiang, J. Emerging uniform Cu2O nanocubes for 251st harmonic ultrashort pulse generation. J. Mater. Chem. C. 2020, 8, 14386–14392. DOI:10.1039/d0tc03622f

We tried our best to improve the manuscript and made some changes in the manuscript. We appreciate for Editors/Reviewers’ warm work earnestly, and hope that the correction will meet with approval.

Once again, thank you very much for your comments and suggestions.

Supporting Infromation for reviewers only

Table S1. List of Nano-Device Information.

device

without annealing

80℃

100℃

150℃

1

CH47-1-M34

ambipolar

ambipolar

n-type

2

CH47-1-H32

ambipolar

ambipolar

n-type

3

CH47-2-H31

ambipolar

ambipolar

n-type

4

CH47-2-H32

ambipolar

ambipolar

n-type

5

CH47-2-H33

ambipolar

ambipolar

n-type

6

CH47-2-H34

ambipolar

ambipolar

n-type

7

CH43-H43

ambipolar

n-type

8

CH43-M32

ambipolar

n-type

9

CH43-M34

ambipolar

n-type

10

CH43-M42

ambipolar

n-type

11

TWE577-1-21

ambipolar

ambipolar

n-type

12

TWE577-1-43

ambipolar

ambipolar

ambipolar

n-type

13

TWE577-1-24

ambipolar

ambipolar

n-type

14

TWE577-4-32

ambipolar

n-type

15

TWE-557-5-22

ambipolar

n-type

16

DM9-22

ambipolar

n-type

17

DM9-23

ambipolar

n-type

18

DM7-11

ambipolar

ambipolar

n-type

19

DM7-22

ambipolar

ambipolar

n-type

20

CH33-13

ambipolar

n-type

n-type

21

CH33-21

ambipolar

ambipolar

n-type

22

CH33-22

ambipolar

n-type

Reviewer 3 Report

Authors report C70/Fc nanosheet field-effect transistor and its conversion from ambipolar to unipolar behavior through thermal annealing.

Authors already published similar result (ref 39) using C60/Fc nanosheet. 

Furthermore, C70/Fc nanosheet is not developed by the authors of this manuscript (it was reported by Osonoe et al (ref 40))

So, this manuscript seems to be lack of novelty. 

Therefore, I cannot recommend the publication of this manuscript.

Author Response

Ms. Ref. No.:  nanomaterials-2532676

Title: Ambipolar to unipolar conversion in C70/Ferrocene nanosheet field-effect transistors

Response to the Reviewers’ comments

Dear Reviewers,

We are very grateful for your constructive comments to our manuscript. The manuscript was revised in accordance with all your comments and queries, which are valuable in improving the quality of our manuscript. All changes made to the manuscript are marked in red. We hope that the new version of our manuscript meets the high standards for publication in Nanomaterials.

Sincerely, on behalf of all authors

Dr. Dorra Mahdaoui

Laboratory of Materials, Molecules and Applications,

Preparatory Institute for Scientific and Technical Studies,Tunisia.

Phone number: 00216 52 30 91 45

Email address: dorra.mahdaoui@ipest.ucar.tn

Reviewers ‘comments:

Reviewer #3:

Authors report C70/Fc nanosheet field-effect transistor and its conversion from ambipolar to unipolar behavior through thermal annealing.

Authors already published similar result (ref 39) using C60/Fc nanosheet.

Furthermore, C70/Fc nanosheet is not developed by the authors of this manuscript (it was reported by Osonoe et al (ref 40))

So, this manuscript seems to be lack of novelty.

Therefore, I cannot recommend the publication of this manuscript.

Answer:

Our group has been studying on "fullerene hybrid nanomaterials" for many years. As part of this, we reported synthesis of C60/Fc nanosheets (2009, ref.34) as a first fullerene hybrid nanomaterial. After that, we developed the research on the electrical and optical properties of these nanomaterials (C60/Porphyrin nanosheets; 2012, ref.37). I also read with great interest Osonoe's research published in 2014, but this time we present the new results obtained from a slightly different perspective in our laboratory.

We tried our best to improve the manuscript and made some changes in the manuscript. We appreciate for Editors/Reviewers’ warm work earnestly, and hope that the correction will meet with approval.

Once again, thank you very much for your comments and suggestions.